# Teaching-learning in clinical education based on epistemological orientations: A multi-method study

**Hamed Khani**[1]*, **Soleiman Ahmady**[1], **Babak Sabet**[2], **Ali Namaki**[1], **Shirdel Zandi**[3], **Somayeh Niakan**[4]

**1** Department of Medical Education, Virtual School of Medical Education & Management, Shahid Beheshti University of Medical Sciences, Tehran, Iran, **2** Department of Surgery, School of Medicine, Shahid Beheshti University of Medical Sciences, Tehran, Iran, **3** Department of Operating Room, School of Paramedicine, Hamadan University of Medical Sciences, Hamadan, Iran, **4** Department of Prosthodontics, School of Dentistry, Tehran University of Medical Sciences. Tehran, Iran

* hkhani95@sbmu.ac.ir; hkhani83@yahoo.com

**Data Availability Statement:** Data and materials availability statement The article and its S1, S2 and S3 Files contain all relevant and underlying data.

## Abstract

### Introduction

Teaching-learning is the heart of medical education in the clinical setting. The objective of this research was to develop a conceptual model of effective clinical teaching in undergraduate medical education and conceptualize its operational framework based on the best fit approach.

### Materials and methods

This research consisted of three sub-studies conducted using a multi-method approach. The first sub-study was conducted using a qualitative meta-synthesis approach. The second sub-study used Clarke's situational analysis approach as a postmodern version of grounded theory. Finally, the third sub-study was designed in two stages. First, it was conducted using the expert panel, in the second step, framework of synthesis based on best fit, and the framework of Ottenhoff- de Jonge et al., which formed the basis of this study.

### Results

In the first sub-study, qualitative evidence on the factors of effective teaching-learning in clinical education was synthesized into five dimensions. Based on the second sub-study, the clinical teaching-learning situation in undergraduate medical education in Iran was represented in three maps, including situational, social worlds/arenas, and positional. Finally, in the third sub-study, based on model modification and development in the expert panel, the effective teaching-learning dimensions were developed into behavioral, social, pedagogical, technology, contextual, educational leadership, and financial dimensions. In the second step, based on the framework of Ottenhoff- de Jonge et al., a three-dimensional matrix was developed concerning epistemological orientations about teaching and learning.

**Funding:** The authors received no specific funding for this work.

**Competing interests:** The authors have declared that no competing interests exist.

## Discussion

Moving from a single teaching-centered and learning-centered orientation to a teaching-learning-centered orientation is required for effective teaching-learning in clinical medical education.

## Introduction

The clinical environment can be defined as a situation with the presence of a medical educator, medical students, clinical staff, and patients revolving around the diagnosis, treatment, and care of the patients and teaching-learning activities. This includes two important points: First, the clinical environment consists of inpatient, outpatient, and community settings [1]. Second, teaching in clinical settings is often done in routine clinical care where patients and their problems are the basis of teaching to medical students [2]. A wide range of professional skills such as communication skills, professionalism, history taking, and physical examination required for medical practice are taught in these environments and settings. But what matters is the effectiveness of teaching and students' achievement of clinical learning outcomes. In other words, clinical teaching-learning should incorporate the components and characteristics that contribute to students achieving learning outcomes. Achieving effective teaching in any educational environment requires the formation of an efficient and high-quality teaching-learning process or system.

Higher education teaching-learning consists of various components and aspects that can be analyzed holistically within the framework of an efficient behavioral system. As a system model, these components and dimensions can be used to create the teaching-learning system in the context of a strategic approach and with appropriate leadership [3].

Comprehensive studies have not been conducted on the dimensions of effective teaching in clinical education. For example, a study focused on successful clinical education and considered components such as the tutor's role, the student's role, the patient's roles and characteristics, and the characteristics of a good clinical environment [4]. Moreover, the perceptions of clinical teachers and students of effective opportunities to facilitate learning in a clinical context have been considered [5]. In another study, Ross & Stenfors-Hayes [6] developed a framework for teaching and learning. This framework includes teacher and teaching activities, learner and learning activities, shared activities between teachers and learners, teaching and learning situations, and content. They propose this framework for the undergraduate course and do not consider the distinction between preclinical and clinical contexts.

Teachers' belief orientation about teaching and learning should be considered to develop the framework of effective educational behaviors. "Beliefs" refer to perceptions and conceptions about teaching and learning, which are formed throughout life, deeply rooted, and contain cognitive and affective aspects [7,8]. The body of research shows that epistemological beliefs are one of the driving forces behind educators' educational behaviors and their choices in teaching design and delivery. In other words, how teachers teach and even teaching practice is unconsciously influenced by their conceptions and perceptions of teaching and learning [9–16].

In recent decades, studies have been published that have conceptualized teaching and learning in education, higher education, and medical education based on a framework. These studies have classified the belief orientation about teaching and learning on a continuum from teaching-centered to learning-centered [3,9,12,15–25]. One of these frameworks proposed for the context of medical education is the framework of Ottenhoff- de Jonge et al., [22]. They

developed a new framework for the context of medical education based on the Samuelowicz and Bain framework [12], which focused on medical educators' beliefs about teaching, learning, and knowledge. Their proposed framework is comprised of a two-dimensional matrix in belief orientations (including six belief orientations: imparting information, transmitting structured knowledge, facilitating understanding, helping the student develop expertise, sharing the responsibility for developing expertise, and negotiating meaning) and nine belief dimensions (including desired learning outcomes, expected use of knowledge, responsibility for transforming knowledge, nature of knowledge, students' existing conceptions, teacher-student interaction, creation of a conducive educational environment, professional development and student motivation). We selected this framework for the following three reasons. First, this framework was largely matched with the subject of our research and therefore had the best fit. Second, this framework is the most comprehensive of all the frameworks developed in the literature. Third, this framework has been developed for the context of medical education.

Overall, this line of research is significant. Because it has implications for teaching and learning activities and contributes to teaching-learning theory, in other words, classifications related to the orientation of teaching-learning are essential and determine the direction of the teaching-learning system. By focusing on its dimensions, the quality of education can be improved. However, these categorizations are limited to teaching-learning in medical education, especially clinical education. Therefore, this study aimed at conceptualizing effective clinical teaching-learning in undergraduate medical education based on epistemological orientations about teaching-learning. Therefore, the objective of this research was to develop a conceptual model of effective clinical teaching in undergraduate medical education and conceptualize its operational framework based on the best fit approach.

## Design of the study

While mixed method research design refers to combining at least one quantitative and one qualitative method, the multi-method research design approach is not limited solely (and entirely) to combining quantitative and qualitative methods. Multi-method research design integrates several quantitative methods or several qualitative methods, or a combination of both methods [26] to make a richer understanding about the subject under study. Tashakkori and Teddlie [27] define the multi-method research as a type of research design in which multiple methods or worldviews are utilized to collect data. This research comprised three sub-studies that conducted using several methods (each of the methods performed rigorously and complete in itself, in one project) the results were then triangulated to form a complete whole.

## Setting and participants

This study, including a wide variety of participants, focuses on the clinical environment of undergraduate medical education (UME). Different participants explained a wide range of perspectives regarding the research topic. In addition, we purposefully selected both groups of experienced clinical teachers and medical education specialists (educational specialists not involved in clinical practice) to address potential differences in beliefs about teaching and learning that may result from engaging in clinical practice. In general, experienced clinical teachers, medical education specialists, and students were present in the second sub-study. Thirty-eight people were recruited for the second sub-study, and 31 participated in web-based interviews. Of these thirty-one people, twenty-two were fifth, sixth, and seventh-year students practically engaged in clinical education (internship), and nine were medical education specialists and clinical teachers. The students were from the Tehran and Shahid Beheshti Universities of Medical Sciences. Fourteen students were male, and eight were female. Finally,

telephone interviews were conducted with seven medical education specialists and clinical teachers. Of the sixteen medical education experts and clinical teachers participating in web-based and telephone interviews, nine were male, and seven were female. The participants of the third sub-study were ten people who were organized as an expert panel. Six were medical education specialists (three male and three female), and four experienced clinical teachers (three male and one female). The clinical professors and medical education specialists who participated in this research were from Tehran, Shahid Beheshti, Shiraz, Guilan, and Ahvaz Jondishapur Universities of medical sciences. The Inclusion criteria for medical education specialists included having a Ph.D. in medical education from Iran or abroad, being a faculty member with academic rank (Assistant professor, Associate professor, and Full professor), and at least ten years of work experience in the field and practice of medical education. The inclusion criteria for clinical teachers included expertise in one of the clinical disciplines, faculty membership, at least fifteen years of teaching experience, and academic rank (Assistant Professor, Associate Professor, and Full Professor). We developed these criteria to locate faculty members whose knowledge and experiences would be enlightening.

## Methods and procedures

This research includes three sub-studies, each of which was conducted using different methods. This research data were collected through multiple methods such as synthesis of qualitative studies, literature review, analysis of the general medicine curriculum and related documents, remote semi-structured interviews (web-based and telephone), expert panel, and the framework synthesis based on the best fit. In addition to documents and curriculum analysis, experienced clinical teachers, medical education specialists, and eligible medical students participated in this study. This study lasted from January 2021 to April 2022.

### First sub-study: Systematic review of qualitative studies and meta-synthesis

The purpose of the first sub-study was to develop a comprehensive framework of effective teaching-learning factors in clinical education. This sub-study was performed based on qualitative meta-synthesis and the seven-step method of Sandelowski & Barroso [28] (Fig 1).

The studies' identification, screening, and selection were performed according to the PRISMA flowchart instructions. PICOS strategy was used to formulate the research question and determine the criteria for eligible studies. In other words, if the research question is well-formulated in review studies, it can guide the determination of eligibility criteria, the search for studies, the collection of data from included studies, and the presentation of findings. PICO is the most common structure used to define a researchable question. Health researchers and professionals extensively employ it to develop searchable questions which provide relevant and accurate results. The PICO strategy ensures that the research question drives the entire review process.

The PICO framework requires the researcher to design the research question based on the following four elements [29]:

P: Population (Patient or the problem to be addressed)
I: Intervention (Exposure to be considered–treatments/tests)
C: Control (Control or comparison intervention treatment/placebo/standard of care)
O: Outcome (Outcome of interest)

As the PICO tool does not currently accommodate terms related to qualitative research or specific qualitative designs, this framework for searching and synthesizing qualitative evidence has been modified to PICOS, where the "S" refers to the study design [30], thus limiting the number of irrelevant articles.

| 1 | **Formulating the research question** |
| 2 | **Conducting a systematic search** |
| 3 | **Screening and selecting appropriate qualitative studies** |
| 4 | **Critical appraisal of studies and extracting the required data from included studies** |
| 5 | **Analyzing and synthesizing of findings of qualitative studies** |
| 6 | **Maintaining quality control (Trustworthiness)** |
| 7 | **Presenting results (conceptual framework)** |

**Fig 1. The seven-step method of meta-synthesis.**

The determined electronic databases and journals were searched from 1990 to 2021 to identify related studies and articles. In order to increase the reliability of the search, this process was carried out by two researchers with the help of a librarian. Based on the composition and search strategies in the databases (OVID, PubMed, Web of Science, SCOPUS, Eric, Magiran, and SID), 33,799 and 56 studies were identified from other sources. After removing duplicates, 29,285 studies and articles remained. Study screening was performed based on inclusion and exclusion criteria by two authors (Hamed Khani and Soleiman Ahmady). After initial screening and assessment based on the abstract, 120 studies and articles remained. The eligibility assessment (full-text assessment) was conducted by two researchers simultaneously and independently. At this step, any disagreement between the researchers was discussed and resolved. In the event of significant inconsistencies, we sought assistance from other research team members or an expert outside the research team. After monitoring and reviewing the full text of the publications, 53 studies and articles were selected, and 45 studies and articles were chosen and included in the meta-synthesis process after a critical assessment of candidate studies using the CASP tool. In order to analyze and synthesize the findings of qualitative studies, inductive coding (reading and reading carefully studies, open, axial and selective coding and production of analytical themes) were used. To ensure the data's trustworthiness, Maxwell [31,32] criteria, such as descriptive, Interpretive, and theoretical validity, as well as Kvale [33] criteria, such as pragmatic validity, were considered in this sub-study.

Descriptive validity refers to the factual accuracy of data [31,32]. In meta-synthesis studies, this type of validation means identifying all relevant research reports and a detailed description of each study's report [28]. Utilizing all channels to search for studies, consulting with librarians and information science experts, contacting the authors of the studies included in the meta-synthesis to resolve any possible ambiguities, utilizing reference management software, and maintaining the audit trail or documentation of all research steps and processes were employed to increase descriptive validity. Interpretive validity is the complete and fair representation of the meanings attributed to the phenomenon under study by the participants [31,32]. In meta-synthesis studies, the actors of the study are the researchers whose study report is included in your project. They are not the participants who were the study subjects

[28]. It is necessary to hold regular meetings to increase the interpretive validity of strategies, such as meetings to discuss the evaluation strategies of the study report. The critical evaluation of the study report under the supervision of the supervisor, two internal and external reviewers, and two peer students, contacting the authors of the studies included in the meta-synthesis to resolve any potential ambiguities, and maintaining the audit trail or documentation of all research steps and processes were employed.

Theoretical validity refers to the theoretical constructs the researcher deals with during the study [31,32]. The primary data in a meta-synthesis study consist of the findings of the studies included in your project. Accordingly, theoretical validity in meta-synthesis studies depends on the credibility of the researcher's methods to develop the integration and researcher interpretation of the researchers' findings of the included studies. To increase the theoretical validity, strategies such as consulting with the supervisor and a number of experts in qualitative methodology and meta-synthesis, as well as maintaining the audit trail or documenting all research steps and processes, were employed.

Pragmatic validity refers to the utility and transferability of knowledge [33]. In meta-synthesis studies, pragmatic validity refers to applicability, timeliness, and translatability to synthesizing evidence and the researcher's product [28]. Strategies such as expert peer review, maintaining the audit trail, and documenting all research procedures were implemented to increase the pragmatic validity.

## Second sub-study: The situational analysis of teaching-learning in clinical education

Clarke's situational analysis approach [34] was used as a post-structural version of grounded theory in this sub-study. The purpose of this sub-study was to represent the fundamental elements and components of clinical teaching-learning in undergraduate medical education in Iran, focusing on identifying the challenges of effective teaching-learning. The data of this sub-study were collected using several methods and sources such as a mini literature review, an analysis of the general medicine curriculum and related documents, and remote qualitative interviews (web-based and telephone). The participants of this sub-study were purposefully selected and entered the research process through the methods of maximum variation (experienced clinical teachers, medical education specialists and students), snowball (experienced clinical teachers and medical education specialists) and convenience sampling (documents and curriculum analysis). In this sub-study, thirty-one people (including experienced clinical teachers, medical education specialists and students) responded to the web-based interviews out of the invited forty. Also, seven people (including experienced clinical teachers and medical education specialists) participated in the telephone interviews. The interview questions included an introductory question to establish rapport with the interviewees and five main open-ended questions (S1 File). These questions were formulated based on Clark's situation analysis literature and theoretical foundations. In other words, to develop these questions, situational, social worlds/arenas, and positional maps were used as the main strategies of situational analysis. In addition, the analyzed content of documents and curriculum were also used to create these questions. The supervisor, two qualitative research experts, and two peer students evaluated the validity and relevance of the questions. The necessary changes were made based on their feedback, and the final questions were developed.

Web-based and telephone interviews questions were the same, with the difference that in the telephone interviews additional questions were developed during the conversations. However, many medical educators, clinical teachers, and students received web-based interview questions via Porsline and answered only the questions included. The web-based interviews

were used to acquire a broad understanding of the situation, followed by the telephone interviews (video and voice call) to gain a rich and deep insight into the people involved.

The average time to answer qualitative interviews (web-based and telephone) lasted about 30 minutes (between 15 to 40 minutes). The notes of the documents, curriculum analysis, and the transcripts of the qualitative interviews were analyzed for the emergence and categorization of sub-themes and themes. Finally, the themes from all data sources were combined to form a comprehensive picture of the situation, represented in three maps. The researchers used three maps (situational, social worlds/arenas, and positional) as the main strategies for situational analysis throughout the research project (from design to reporting). Lincoln and Guba's criteria [35] were used to increase the rigor of the data in this study. Memoing, prolonged engagement with data, member checking, peer checking, coding and categorization of the emerging themes by the researchers, and establishing a consensus were all employed to assure the credibility and dependability of the findings. To guarantee the confirmability of strategies such as devoting sufficient time to data collection and analysis, utmost accuracy in the research process and audit trail were used. Finally, to ensure the transferability of strategies such as the thick description of the results in the form of discussion about the findings, quality assessment of data by two medical education specialists and experienced clinical teachers and different participants in terms of position were used.

## Third sub-study: Expert panel and best fit approach

The purpose of this sub-study was to present a conceptual model of effective teaching in clinical education and conceptualize its operational framework based on epistemological orientations about teaching and learning. This sub-study was conducted using a qualitative approach (expert panel method and best-fit approach).

**Expert panel method.** The expert panel method is the forum in which prominent and expert people are invited to express and share their experiences, thoughts, and ideas' in a particular field [36]. This method is often based on the modified Delphi structure [37].

In this sub-study, an expert panel was used, which included two phases question-centered and discussion (S2 File). In the first step, in order to collect data, before forming a group discussion in the expert panel, the following two questions were sent to 12 experts via Porsline (web-based):

How do you define effective teaching-learning in clinical education?

What components and elements should be included in clinical teaching-learning in undergraduate medical education to enable students achieve the goals and outcomes of clinical learning?

After collecting the responses to these questions, and implementing and assessing them, a 10-member expert panel was constituted (6 medical education specialists and four clinical teachers). This panel was held virtually through the Skype platform in two rounds (Each round in 2 hours). In addition to these two rounds, other data were collected through the WhatsApp group in an unstructured manner. In the first round, a brief introduction was given, and the members' and research team's expectations were stated to develop rapport between the person in charge of the panel and the members and between members. The head of the panel (the research team's leader) then presented and discussed the first and second sub-studies (purpose, design, and findings) to the members, answering questions and leading a group discussion. In the next step, the second round of the expert panel was held with an interval of one week and focused on developing and modifying the model obtained from the first sub-study based on the components and elements obtained from the situational analysis (second sub-study). At this stage, the interview guide (discussion) was used. However, before

the second round and with an interval of one week between the first and second rounds, the findings and results of both sub-studies were shared in a schematic to facilitate a logical and objective consensus among the panel members form in the WhatsApp group. Eight open-ended questions (S3 File) were sent privately over WhatsApp in order to framework synthesis, and participants were requested to return the responses in the form of a recorded voice. This aimed to achieve an overview of their epistemological orientation in relation to teaching and learning. The participants of this sub-study entered the research process purposefully through reputational case sampling. The data from this sub-study was recorded, implemented, and coded at each step, and sub-categories and categories were classified using direction content analysis. The rigor and trustworthiness of the data in this sub-study depended entirely on guaranteeing the trustworthiness of the data in the first and second sub-studies. In addition, the final model was examined and validated in this step by the supervisor and expert peer review (two external experts who are specialists in the field) and their feedback was used.

**Framework synthesis based on the best fit.** Framework synthesis is one of the methods developed for synthesizing qualitative data, which is mainly a deductive approach [38]. The "best fit" framework synthesis method is an approach and mean for testing, reinforcing, and developing an existing published model or framework, which is presented for a potentially different but relevant population (same context) [39]. Framework synthesis based on best fit requires identifying a framework, theory, or conceptual model related to the research subject. Following steps such as a systematic review of qualitative studies and meta-synthesis (first sub-study), situational analysis (second sub-study), and model development based on the first and second sub-studies using an expert panel, which was somewhat in line with the method proposed by Carroll et al., [40] the authors focused on reviewing existing published frameworks and models of teaching-learning in the fields of education, higher education, and medical education in this phase. The framework synthesis approach was used after they discovered a published framework in the literature that conceptualized teaching-learning in the practice of medical education [22]. Although this framework did not fully match the research subject, it had the best fit. Finally, an operational framework was conceptualized based on the analysis of participants' responses to open-ended questions in relation to their beliefs about teaching and learning and using the best fit approach.

Two sets of inclusion criteria and a search and selection of studies and articles are required in the best-fit approach. The first is for a systematic review of qualitative studies conducted in the first sub-study, and the second is for searching and identifying a related model or framework. These criteria, which are consistent with the PICO strategy (population, intervention, comparison, outcome) and the SPICE strategy (setting, perspective, intervention/phenomena of interest, comparison, and evaluation), are presented in Table 1 for both systematic reviews of qualitative studies and the identification of related model and framework. The reliability of the search was ensured by utilizing these two strategies.

## Ethical approval and consent to participate

This article is taken from the Ph.D. dissertation of Dr. Hamed Khani from the Department of Medical Education of Shahid Beheshti University of Medical Sciences and has received ethics approval with the number IR.SBMU.SME.REC.1399.097 on 2021-01-13 from the university's ethics committee. Verbal and written consent was obtained from all participants to participate in telephone interviews (voice recording), web-based interviews, and expert panel. All participants were informed of the research objectives at the time of data collection and were assured that participation in qualitative interviews (web-based and telephone) and expert panel was voluntary. Data confidentiality and anonymity of participants were guaranteed in the

Table 1.  Inclusion criteria for systematic review of qualitative studies and identification of related framework.

| | Models/ frameworks and theories | Primary qualitative studies (first sub-study) |
|---|---|---|
| Setting/ population | Undergraduate Medical Education/ students/ educators / patients | Clinical education in undergraduate medical/ Clinical training in undergraduate medical/ students / educators / patients |
| Intervention/ phenomena of interest | Teaching-learning in higher education and medical education | Effective teaching-learning in clinical education |
| Research design/ evaluation | Existing published frameworks and models on teaching-learning in medical education | Qualitative approach, grounded theory, ethnography, phenomenology, thematic analysis, content analysis, Delphi method, focus groups and discourse analysis |

qualitative interviews, and expert panel, both verbally and writing. Finally, all participants were informed that will be presented to them the research results if they request. All methods were conducted in accordance with the ethical principles of the Declaration of Helsinki. The Declaration of Helsinki is one of the most important international documents on ethics in research. The World Medical Association (WMA) has developed the Declaration of Helsinki as a statement of ethical principles for medical research involving human subjects, including research on identifiable human materials and data. The first version of this declaration was adopted by the 18th Assembly of the World Medical Association in Helsinki, Finland, in June 1964. Since then, it has been revised seven times, with the most recent update occurring in 2013 [41]. Subject classification and clauses of the latest version of this declaration [41] include 12 subjects and 37 clauses. In general, the basic principles of this declaration include which protection of life, health, dignity, integrity, right to self-determination, privacy, and confidentiality of personal information of research subjects is the duty of physicians and researchers involved in medical research [41,42].

## Results

### Findings of first sub-study

According to the findings of the first sub-study, seven components from the included qualitative research were synthesized into five primary dimensions, and a useful teaching-learning framework for clinical education was developed. This framework includes dimensions such as behavioral or content (learner, teacher, patient and her/his behavior), social (collaborative learning community), pedagogical (instructional design and teaching-learning opportunities), the context of teaching-learning (positive and supportive clinical environment) and educational leadership (classroom management and structure) (see Fig 2) [43].

### Findings of second sub-study

Based on the results of this sub-study, clinical teaching-learning in undergraduate medical education in Iran was represented in three maps. The first map formulated human, non-human, material, symbolic and discursive components and elements for teaching-learning. Then, the discourses and themes that emerged in the messy version were organized and specifically represented using the ordered version of the situational map. The second map represented social worlds/arenas of teaching-learning situations in the clinical training in undergraduate medical education in Iran. According to this map, the three main arenas such as a hospital (including ambulatory education and training in the outpatient clinics, education in hospital wards, inpatient settings, bedside teaching and clinical skills learning centers), universities and medical schools (including the clinical skills learning center and classrooms), and health centers arena/ comprehensive health services (including outpatient education and

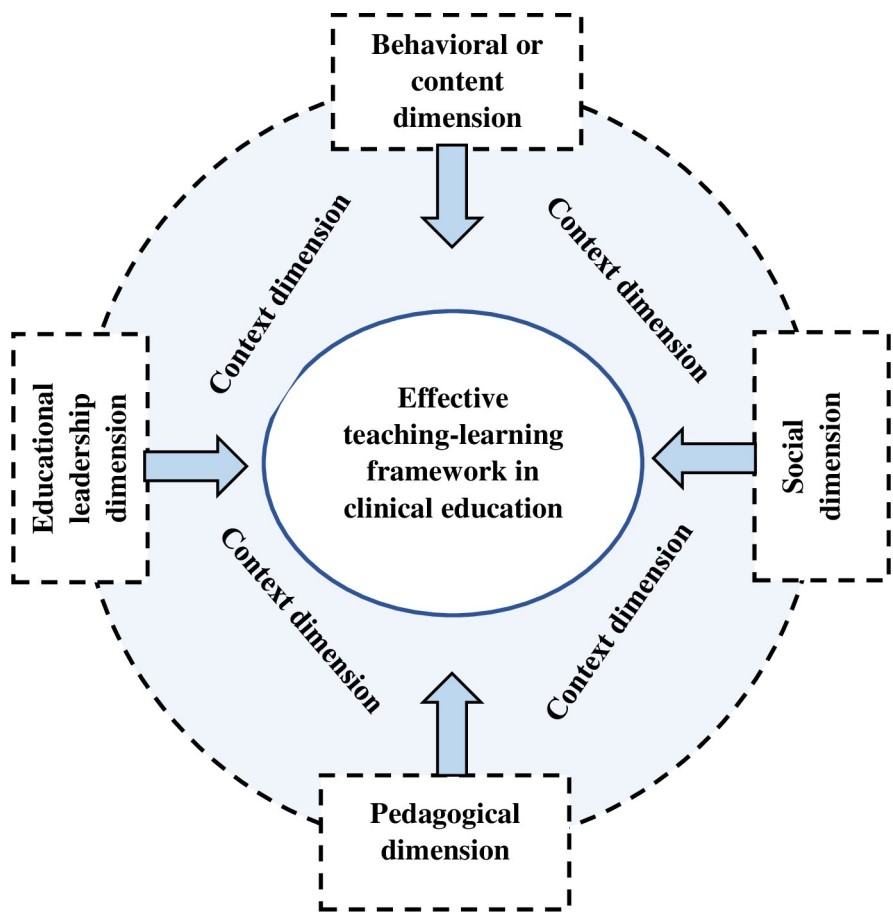

**Fig 2. Results of qualitative meta-synthesis in relation to the components and dimensions of effective teaching-learning in clinical education.**

slightly community-based education) were identified for clinical education. In addition, in the second map, the worlds and social discourses residing in these arenas include (clinical teachers and educators, students, hospital physicians, residents, nurses, non-educational staff, family physicians, and simulation and technology) were represented. According to the third map (positional map), the challenges and problems related to clinical training in undergraduate medical education in Iran were illustrated in six positions, which include challenges and problems of curriculum (position 1), challenges and problems related to culture, behavior and attitude in clinical education (position 2), challenges and problems of management and leadership in clinical education (position 3), challenges and problems related to the environment, space and time in clinical education (position 4), challenges and problems of financial in clinical education (position 5) and challenges and problems related to equipment and technology in clinical education (position 6). Finally, elements and recommendations were provided based on this map to develop and support effective clinical teaching [44].

### Findings of third sub-study

**Expert panel findings.** Ten experts (6 medical education specialists and four experienced clinical teachers) were present in the expert panel. Descriptive findings of them is provided in Table 2.

**Table 2. Number and distribution of expert panel according to gender, field of expertise and academic rank.**

| Descriptive statistics | | N | Percentage |
|---|---|---|---|
| **Panel members in terms of medical and clinical education** | | | |
| Medical education specialists | | 6 | % 60 |
| Experienced clinical teachers | | 4 | % 40 |
| **Gender** | | | |
| Medical education specialists | Male | 3 | % 50 |
| | Female | 3 | % 50 |
| Experienced clinical teachers | Male | 3 | % 75 |
| | Female | 1 | % 25 |
| **Specialized field and teaching practice** | | | |
| Medical education specialists | Clinical education | 2 | 33/3 |
| | Curriculum planning and program evaluation | 1 | 16/7 |
| | Teaching-learning theories | 2 | 33/3 |
| | Student assessment | 1 | 16/7 |
| Experienced clinical teachers | Gastroenterology | 1 | % 25 |
| | Social medicine | 1 | % 25 |
| | Pediatrics | 1 | % 25 |
| | Surgery | 1 | % 25 |
| **Academic rank** | | | |
| Medical education specialists | Assistant Professor | 3 | % 50 |
| | Associate Professor | 3 | % 50 |
| | Full Professor | - | - |
| Experienced clinical teachers | Assistant Professor | 1 | % 25 |
| | Associate Professor | 2 | % 50 |
| | Full Professor | 1 | % 25 |

In the first step of the third sub-study, based on the expert panel method, the results of the first and second sub-studies were combined with a deductive approach. In this sub-study, the model obtained from the first sub-study was strengthened and developed based on the elements that emerged in the second sub-study, and an effective clinical teaching-learning model was developed for undergraduate medical education in Iran (see Fig 3).

Based on the expert panel discussions, individuals such as residents, hospital physicians (specialist and attending physicians), nurses, and non-educational staff present in the context of clinical education were considered and included in the final model. According to Clark's methodology, these individuals are on the margin of teaching-learning situations, meaning they do not formally educate students but are involved in teaching-learning in opportunistic situations. Therefore, they assist students in achieving learning goals and outcomes through hidden and informal teaching.

In addition, based on the synthesis of the results of the first and second sub-studies in the expert panel discussion, the main arenas of teaching-learning in the clinical training in undergraduate medical education were considered in the final model. These arenas include hospitals, medical schools and health centers (comprehensive health services). Accordingly, in the arenas of clinical teaching-learning, or more generally in the context of clinical education, different social worlds influence the teaching-learning discourse.

Also, based on the combination of the results of the first and second sub-studies in the expert panel discussion, the component of teaching-learning culture is formed from the intersection of two behavioral dimensions (student, teacher and patient) and social (interaction between student-instructor-patient). Teaching-learning culture is at the heart of the clinical context and as part of this system (clinical environment culture) impacts the students' achievement or non-achievement of clinical learning outcomes. Clinical teaching-learning culture can

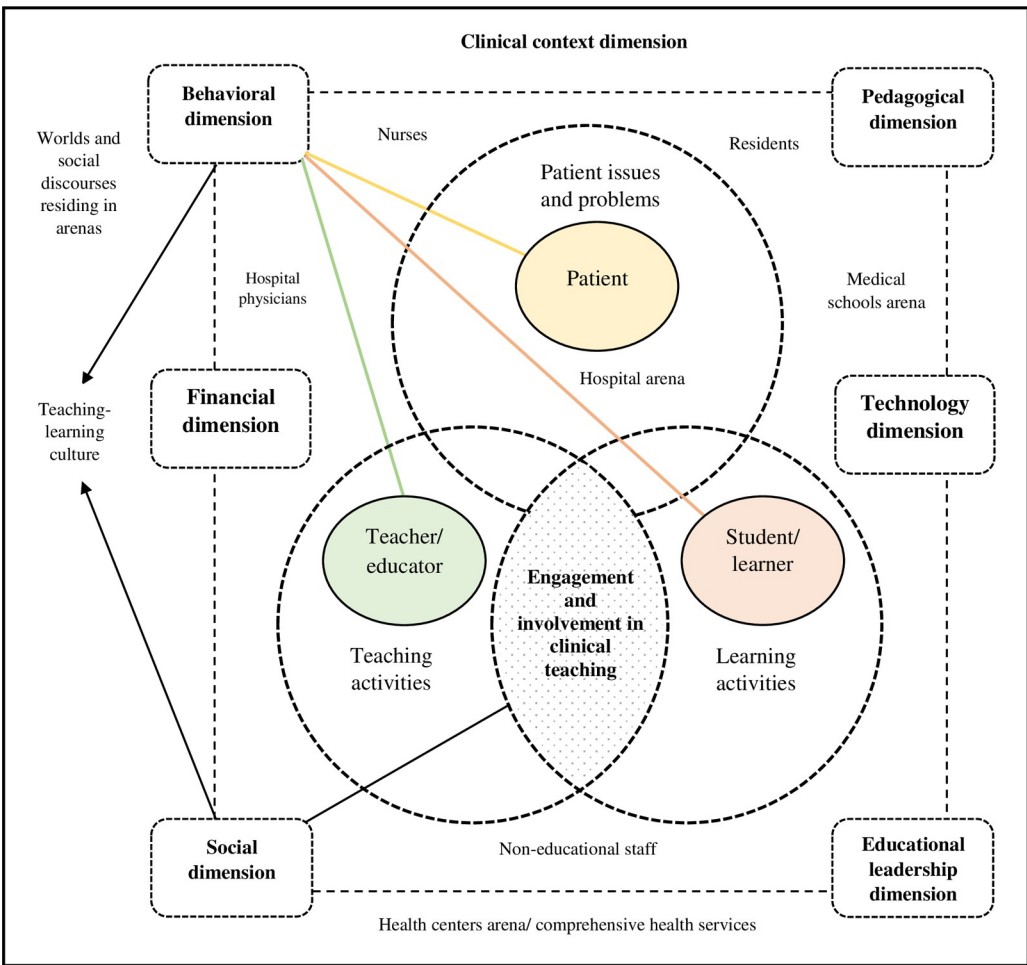

**Fig 3. The final model of effective clinical teaching-learning for undergraduate medical education based on the synthesis of the results of the first and second sub-studies in the expert panel.**

be influenced by residents, hospital physicians (specialist and attending physicians), nurses, non-educational staff, as well as the social worlds of these people.

Moreover, based on the synthesis of the first and second sub-studies in the panel discussion, the financial dimension was added to the final model as a factor that affects effective teaching-learning. Lack of funding for education is one of the obstacles to effective teaching-learning in clinical training in undergraduate medical education in Iran. In other words, by providing financial resources, the quality and effectiveness of clinical education can be increased.

Based on the model's modification and development in the expert panel's discussion, the technology dimension with three subcategories (distance clinical education, simulation and technologies-enhanced learning) was added to the final model.

Unlike the results of the synthesis of qualitative studies (the first sub-study), which identified and categorized teachers' knowledge and skill in information and communications technology as one of the components of an effective teaching-learning framework, this sub-study, based on expert panel discussion, presented a more developed perspective of technology in clinical education and technology in clinical education was considered as one of the components of an effective teaching-learning framework. This importance and emphasis was the effect of COVID-19 on medical education.

In fact, most of the discussion in the expert panel was about clinical education during the COVID-19 era and after the post-pandemic. This critique noted the model obtained from the first sub-study, which did not consider the technology dimension alongside other dimensions. This critique noted the model obtained from the second sub-study, which did not consider the technology dimension alongside other dimensions. Finally, the PCC-Best conceptual model was developed, which includes seven dimensions of pedagogical, context, content, budget, educational leadership, social, and technological, and can be used to conceptualize, design, and organize clinical teaching-learning in undergraduate medical education in Iran. The naming of the conceptual model obtained from this research is based on the acronym of seven dimensions, which is presented below.

**P** (Pedagogical) **C** (Context) **C** (Content)—**B** (Budget) **e** (Educational leadership) **s** (Social) **t** (Technology)

In the next step, the framework synthesis was performed based on the best fit, and an operational framework was conceptualized for the proposed model in relation to effective teaching-learning in clinical education. The results are presented below.

**Framework synthesis based on the best fit.** The framework synthesis based on best fit requires identifying a framework relevant to the research subject. The framework of Ottenhoff- de Jonge et al. is one of the frameworks that conceptualized teaching and learning in medical education [22]. Because this framework best fits the current study, it served as the foundation for data analysis in this section of the study. The analysis of the experts' answers to the eight open-ended questions about teaching and learning was relevant and in line with the original framework. We preserved the option of changing the dimensions of the primary framework open based on prospective expert opinions. The experts' responses, which were short recorded audios, were first transcribed on paper, then read and re-read to identify the areas of meaning that reflected participants' orientation and beliefs. In this way, their understanding and conceptualization in relation to teaching and learning and the meaning of the dimensions of the research model were labeled according to the dimensions of the framework of Ottenhoff- de Jonge et al., [22]. The fragments extracted from texts which did not cover exclusively one of the dimensions of the original framework or were the common point of both dimensions were added to the primary framework as a new dimension. In contrast to Ottenhoff-de Jonge et al., [22], the new framework incorporates a three-dimensional epistemological orientation matrix on which the seven dimensions of the final model obtained by the expert panel are presented (see Table 3).

Although some components of our framework are common to the framework of Ottenhoff-de Jonge et al., [22], our framework and research data provide a more extended and holistic perspective. Therefore, the new framework (Table 3) consists of three epistemological orientations of teaching and learning. These epistemological orientations include Teaching-centered, learning-centered and teaching-learning centered, which are defined and arranged in a column. This matrix shows the seven dimensions of effective teaching-learning in clinical education in a row. Pedagogical, content or behavioral, social, technological (clinical teaching using new technologies), budget and financial, context, and educational leadership are all included.

As seen in the proposed matrix, each of the seven dimensions has its own meaning in terms of epistemological orientation. For example, the pedagogical dimension in the teaching-centered epistemological orientation is teacher-centered. Meanwhile, it is student-centered in the learning-centered orientation, and the pedagogical dimension in the teaching-learning-centered orientation means teacher facilitation and student-centered. Additionally, critical pedagogy (patient involvement in clinical teaching) is also taken into account. Thus, in this matrix, the meaning of other dimensions is presented in the three epistemological orientations: teaching-centered, learning-centered, and teaching-learning centered. Furthermore, based on this

**Table 3. Dimensions of effective teaching-learning in clinical education based on the range of epistemological orientations.**

| N | Dimensions | Range of epistemological orientations | | |
|---|---|---|---|---|
| | | Teaching-centered orientation | Learning-centered orientation | Teaching-learning centered orientation |
| 1 | **Pedagogical** <br> • Curriculum planning, objectives and desired learning outcomes, teaching approaches and methods, assessment and learning opportunities | Teacher-centered | Student-centered | • Teacher facilitation <br> • Student centered <br> • Critical pedagogy (patient involvement in clinical teaching) |
| 2 | **Content or behavioral** <br> • Positive personality traits of students (motivation, communication skills, love and interest to learning, etc.) <br> • Development of student' autonomy and self-direction <br> • Preparation, knowledge, skills and previous experiences of the student <br> • Positive personality traits of the teacher (interest and enthusiasm, motivation, humor, etc.) <br> • Teacher experience, knowledge and clinical ability <br> • Professionalism and role modeling of clinical teacher <br> • Individual characteristics of patients <br> • Patient problems and their educational value | Teacher' action, work and behavior in the center | Student' action, work and behavior in the center | Teacher, student, patient and triangular behavioral system |
| 3 | **Social** <br> • Teacher-student-patient interaction <br> • Multidisciplinary team-interactive approach in clinical teaching and care <br> • Collaborative teaching-learning <br> • Worlds and social discourses present in the arenas (students, teachers, patients, residents, specialist and attending physicians, nurses and non-educational staff) | One-sided by the teacher | Mutual to maintain student' attention and focus | Network and interactional to negotiate and build a collaborative community |
| 4 | **Technology (clinical teaching based on new technologies)** | for transmitting unstructured information and knowledge | Lack of educational design, for opportunistic learning | Integration in face-to-face clinical training to achieve learning outcomes |
| 5 | **Budget and financial** | Serves to external motivation of the teacher | Not emphasized (not important) | Equipping and strengthening teaching-learning settings in clinical education |
| 6 | **Context** <br> • Creation and promoting of a positive, conducive and non-threatening learning environment <br> • Strong teaching-learning culture <br> • Arenas and worlds and discourses present in them | Not emphasized (not important) | In order to promoting relaxation in the students is somewhat emphasized | Highly important / focus on context as a full- range and comprehensive educational architecture |
| 7 | **Educational leadership** | Autocratic, authoritarian and tyrannical | Delegative or laissez-faire leadership | Democratic and participative leadership |

matrix, effective teaching-learning and achieving learning objectives could not be implemented in terms of the significance of each of these dimensions in both teaching-centered orientation and learning-centered orientation. For example, in teaching-centered orientation, the social dimension is a one-way flow by the teacher, which will not lead to the creation of an effective teaching-learning process. At the same time, in the learning-centered orientation, the application of technology in clinical teaching without educational design can only serve opportunistic learning. Accordingly, in Fig 4, the operational framework for effective teaching-learning entitled (ECT-TLCO) based on the best fit approach was conceptualized.

**E** (Effective) **C** (Clinical) **T** (Teaching)- **T** (Teaching) **L** (Learning) **C** (Centered) **O** (Orientation)

In fact, the meaning of this framework is that effective clinical teaching can be implemented in terms of the epistemological orientation of the teaching-learning centered.

Finally, while Ottenhoff-de Jonge et al's framework [22] served as the basis for the construction of our research framework, the two are similar in some ways and different in others. Table 4 compares the dimensions and components of these two frameworks.

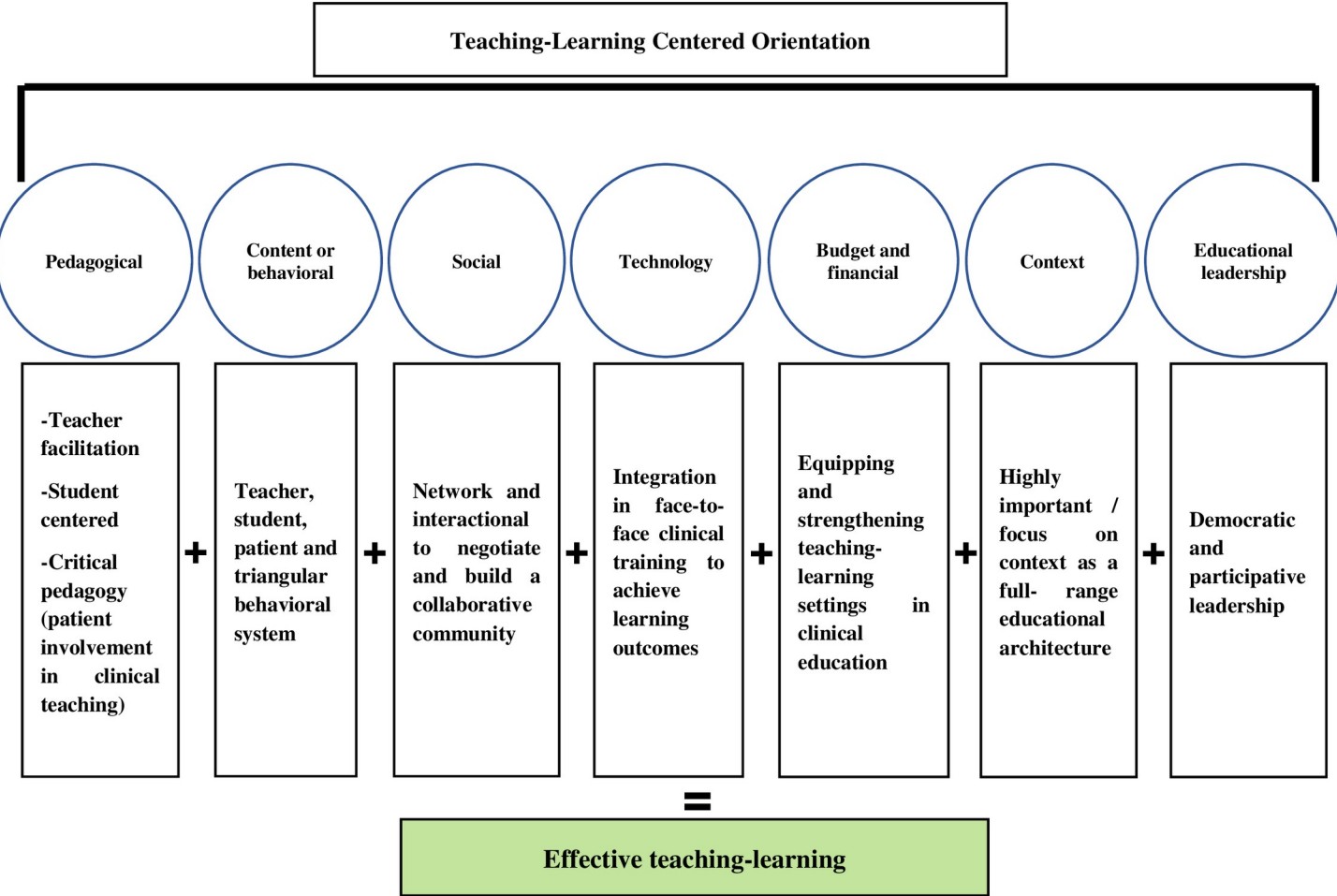

**Fig 4. Conceptualizing the operational framework of effective teaching-learning based on the best fit approach.**

As seen in Table 4, the dimensions and components of the two frameworks are compared. One of the most important differences between the framework of Ottenhoff- de Jonge et al., [22] and the new framework is that, unlike the previous framework, which suggested a two-dimensional matrix, the new framework provides a three-dimensional matrix for epistemological orientations in connection to teaching and learning.

Furthermore, some dimensions of the new framework, such as expected use of knowledge, responsibility for transforming knowledge, and nature of knowledge, are not applicable because Ottenhoff-de Jonge et al., [22], conceptualized medical educators' beliefs about teaching, learning, and knowledge in their study, whereas our study only focuses on teaching-learning. Finally, in the new framework, three dimensions of technology (clinical teaching based on new technologies), budget and financial and educational leadership have been considered, which have not been addressed in the previous framework.

## Discussion

The first sub-study of this research synthesized elements and dimensions of effective teaching-learning in five dimensions based on a systematic review of qualitative studies and meta-synthesis. The second sub-study provided the teaching-learning situation in undergraduate

**Table 4. The comparison of the dimensions of the new framework and the framework of Ottenhoff- de Jonge et al., [22].**

| Dimensions | Comparison of the new framework to the Ottenhoff- de Jonge et al framework |
|---|---|
| **Desired learning outcomes** | This dimension is identical in both frameworks but developed in the new framework as the pedagogical dimension (curriculum planning, objectives and desired learning outcomes, teaching approaches and methods, assessment and learning opportunities). |
| **Expected use of knowledge*** | Not applicable for the new framework |
| **Responsibility for transforming knowledge*** | Not applicable for the new framework |
| **Nature of knowledge*** | Not applicable for the new framework |
| **Students' motivation** and **Students' previous and existing conceptions** and **Students' professional development** | To some extent, these two frameworks have a common language in these components. The previous framework focused on the students' motivation, their previous and existing conceptions and students' professional development, While the new framework considers the behavioral system of students, teachers and patients and has been extended in the content or behavioral dimension (such as students' motivation, preparation, knowledge, skills and previous experiences of the student, development of student' autonomy and self-direction, teacher' motivation, individual characteristics of patients, patient problems and their educational value and etc.) |
| **Teacher-student interaction** | This dimension is identical in both frameworks, but it has been extended in the new framework as the social dimension (including; teacher-student-patient interaction, worlds and social discourses present in the arenas and etc.) |
| **Creation of a conducive learning environment** | This dimension is the same in both frameworks, but it has been developed in the new framework as the context dimension (including; the creation and promotion of a positive, conducive, and non-threatening learning environment, teaching-learning solid culture and arenas and worlds and discourses present in them) |
| **Technology (clinical teaching based on new technologies)**** | This dimension is not considered in the previous framework |
| **Budget and financial**** | This dimension is not considered in the previous framework |
| **Educational leadership**** | This dimension is not considered in the previous framework |

* Not applicable for the new framework
** New.

medical education's clinical training, focusing on its challenges, utilizing Clarke's situational analysis approach. Finally, the conceptual model (PCC-Best) was developed in seven dimensions of pedagogical, context, content, budget, educational leadership, social, and technology in the third sub-study, based on the modification and development of the model in the expert panel, and teaching-learning in the clinical training of undergraduate medical education can be conceptualized, designed, and organized based on it.

Although this model includes elements and dimensions such as pedagogy, content, or behavioral (teacher, student, and patient and their characteristics), context, and environment, it is more complex and developed than previous research results [4,6], and includes other elements and dimensions such as budget, educational leadership, social, and technology for the best and most effective clinical teaching.

Furthermore, the framework synthesis in this study was carried out using one of the available frameworks in the literature [22] that best fit the subject of our research. Finally, based on this approach (best fit), an operational framework entitled (ECT-TLCO) was conceptualized for effective teaching-learning.

Although the new framework of this study confirms some of the old framework's findings, dimensions, and components, it also contains significant differences. One of the important differences between this study and the study of Ottenhoff- de Jonge et al., [22] is related to the study context. They developed their framework, emphasizing the preclinical teaching context, whereas the current study focuses on the clinical context. Another difference between the new framework and the previous one is that the framework of this research offers a more extended perspective of the teaching-learning dimensions. In other words, in addition to conceptual development, in the new framework, there are dimensions such as budget and financial, technology and educational leadership that has not been addressed in the previous framework. Another difference is that some components of the previous framework are not applicable to the new framework; this is because the old framework, in addition to teaching and learning, also considered knowledge belief orientation.

Another important difference between the new framework and the previous one is the shape and structure of the matrix. In the previous framework, the belief orientation was a two-dimensional matrix, while in the new one, the belief and epistemology orientation is three-dimensional. In fact, the previous framework [22] took into account teaching-centered and learning-centered orientations, whereas the new framework also takes into account teaching-learning centered orientation.

Based on previous research, learning-centered and teaching or content-centered orientations are separate categories [45] and have their characteristics. However, the results of other studies indicate that these two orientations are poles of a continuum [12,22,46]. This view defines the teaching or content-centered orientation as "an approach that is not learning-centered. However, the result of our research combines the two opposing viewpoints presented previously. On the one hand, it provides a continuum with three orientations: teaching-centered, learning-centered, and teaching-learning-centered.

On the other hand, in the presented framework, the seven dimensions, such as; pedagogical, context, content, budget, educational leadership, social, and technology, have their characteristics according to the belief orientation, which has intensified the boundary between these orientations. For example, the nature of the social dimension introduced in other research [12,22,47] under the title of student-teacher interaction is different in these three orientations. In teaching-centered orientation, the nature of social dimension and interaction is one-sided by the teacher. In a learning-centered orientation, its nature is mutual to maintain students' attention and focus. In teaching-learning-centered orientation, its nature is network and interactional to negotiate and build a collaborative community. In other words, contrary to the research of Kember & Kwan [47] and consistent with the research of Samuelowicz & Bain [12], Postareff & Lindblom-Ylanne [20], and Ottenhoff-de Jonge et al. s [22], the boundary between the three orientations presented in our research is determined by the nature of the social dimension and interaction, not the interaction and the social dimension per se.

In contrast to what Samuelowicz and Bain [12] assert in their study, the line between teaching-centered and learning-centered orientations is relatively "hard." In our study, however, and following the framework of Ottenhoff-de Jonge et al. [22], the unique characteristics of each orientation's seven dimensions highlight the difference between teaching-centered, learning-centered, and teaching-learning-centered orientations(see Table 3).

Regarding the belief orientation toward teaching, previous research has identified two approaches, teaching-centered (content-focused) and learning-centered [12,16,20,22,47]. But the present study's findings suggest that the theory of teaching approaches should transcend the dichotomy between learning-centered and teaching-centered approaches. And in medical education, to improve teaching quality, the relationship between these two approaches should be emphasized; as a consequence of this complex relationship, the teaching-learning-centered

approach emerges. Theoretically, teaching-learning orientation means teachers should apply the principles and rules of different learning theories, such as constructivism, behaviorism, and cognitivism, to teach effectively. In other words, based on this orientation, educators should reconcile different learning theories such as behaviorism, cognitive, and constructivism in teaching practice.

In addition, we believe that effective teaching-learning may be implemented using this epistemological perspective (teaching-learning oriented) because teaching and learning are two sides of the same coin. In this regard, Thomas Angelo says, "teaching in the absence of learning and without learning is a futile activity." Therefore, the effectiveness of teaching reflects the learning rate of students [48]. In general, with the epistemological orientation of teaching-learning-centered and extensions of seven dimensions in this orientation (such as teacher facilitation, student-centered and critical pedagogy/ patient involvement in clinical teaching, focus on the teacher, student, patient behavioral system, networked social interactions and build a collaborative community, integration of technology in face-to-face clinical training, equipping and strengthening teaching-learning settings, focus on context as a full-scale educational architecture, and democratic and participative leadership), effective teaching-learning can be implemented in clinical education.

In terms of the clinical environment and the extensions of the seven dimensions in the teaching-learning focused orientation, it can be argued that the teaching-learning triangle is formed by the teacher, student, and patient. To create effective teaching-learning, it is essential to focus on these three. Educational policymakers should make this possible by selecting motivated and interested students in the medical profession. Throughout the course, students' self-directed learning skills must be strengthened and their development as lifelong learners. Students' learning in clinical settings and contexts is highly dependent on emotional, educational, and organizational support [49–51], which should not be overlooked.

The teacher is an important part of the educational program [52–55]. Accordingly, teachers and educators have an important role in students' clinical learning. Thus, recruiting competent teachers is significant in medical education, and their personal and professional development must be taken into account during the service.

Contacting real patients plays an essential role in educating students, teachers, and physicians [56–58]. Patients should not be seen as merely "subjects for teaching-learning" Educational policymakers and clinical educators should involve patients in clinical education, curriculum design, or evaluation. In other words, a culture of patient involvement in education must be established, and patients' voices must be heard in the educational process.

Generally, patients prefer participating and being involved in the clinical teaching process. Basically, teaching with patients allows three key domains of learning to be integrated with teaching [59]: A) clinical (knowledge and skills); B) Professional character or professionalism (teamwork and ethical considerations); C) Communications (with staff and patients).

Interaction with some patients is difficult for medical students, especially if the patient is hostile, angry, uncooperative, disinterested, overly talkative, or has chronic pain [60].

Interaction in clinical learning environments is crucial. Interactions in the clinical environment should be considered "Key teaching moments" along with opportunities for tutors to help students develop competence in communication skills [61]. In clinical education environments and settings, the principles of constructivist theories and adult learning can be the basis for teaching-learning. Accordingly, the use of collaborative learning strategies such as small group teaching, problem-based learning, team-based learning, peer learning, etc. can be great mechanisms. In fact, it is only through participation that new methods are learned and new tasks are gradually performed [62]. In addition, in these environments, conditions must be provided for students to build their own knowledge as adult learners.

Contrary to popular belief, leaving learners and students in a clinical setting has no pedagogical basis. It is better for clinical teachers to be equipped with pedagogical knowledge. Pedagogical knowledge is a term used for knowledge of how to teach that can be used in a wide range of educational fields. Therefore, in clinical teaching, educational design and even planning of teaching-learning opportunities are of great importance [63]. In this regard, engaging in faculty development programs effectively develops them.

Regarding the technology dimension, it can be said that although simulations and new technologies such as virtual reality, augmented reality, virtual patient, etc., are increasingly used in health professions education, the long-held tradition of teaching with the engagement of real patients remains valuable [62]. Modern technologies in medical education are important because they have been able to guide opportunistic and informal learning in clinical settings and create a constructive alignment between this type of learning and formal educational activities [64]. Accordingly, the integration of technology in face-to-face clinical education is important.

The context, in general, and the educational environment or atmosphere, in particular, are the other dimensions. Various educational theorists have emphasized the importance of context in education. Michael Allen [65] emphasizes the importance of context in the design of learning interventions. He notes, "In many ways, context is both the most fundamental component of education and often frequently neglected." Colvin Clark [66] refers to the focus on the context in education as an immersive architecture or whole-task instructional design. In general, context is important in medical education, and it can be said that when information is applied to a situation by a person, a dynamic interaction occurs. Ignoring the environment and situation in which knowledge is applied is metaphorically similar to "focusing (only) on the hammer" when nailing to a wall or board. In this state, the nail used, the wall or board to which the nail is affixed, is neglected. The learning environment or "educational atmosphere" as one of the components of context is one of the key aspects of the curriculum that is less tangible than other aspects of the curriculum. According to Genn [67], the "educational climate" is the soul of the medical curriculum. In this regard, clinical teachers, educators, and curriculum planners should consider measuring the educational environment as part of curriculum evaluation and promote an appropriate learning-learning environment. Training within the clinical settings, such as bedside teaching, inpatient education, outpatient clinic, and community education, is at the heart of healthcare education and provides a vital component of clinical education. This training guides students in the clinical environment's culture and social aspects of the clinical environment and shapes students' professional values to prepare them for future work and activity [62,68].

Medical teachers and educators are engaged in a wide range of activities, including teaching-learning, curriculum development, assessment and evaluation, and team and program management. All of these activities need leadership in some way [69]. Therefore, they must be prepared for this important role. In addition, teaching-learning in clinical settings requires participatory leadership. Participation in leadership and management training courses is encouraged in this regard in order to develop them.

Finally, financial support is particularly important in clinical education [70]. Resources to purchase educational materials and technology and not equipping teaching-learning environments in clinical education are important financial constraints and can impair the quality and effectiveness of teaching-learning.

## Limitations

While there may be valid articles, studies, and frameworks in the literature that were not included in this research, searching for articles and studies in the first sub-study and finding the relevant framework and model in the literature to framework synthesis in the third sub-study was based on Persian and English (one of the inclusion criteria). Other limitations of this study were related to semi-structured telephone interviews in the second sub-study. Two participants did not allow their voices to be recorded, and the interviewer was forced to write down the conversation in these two interviews during the interview. Another limitation of this research is related to the context in the third sub-study. The specific cultural differences and characteristics may have influenced epistemological beliefs about teaching-learning in clinical education. For this reason, caution should be taken in transferring this epistemological and classified framework to the context of other countries. Finally, some experts did not respond to the question-based phase questions submitted to them via the Porsline (web-based) due to their busy schedules. As a result of this lack of cooperation, they also did not participate in the rounds of the expert panel.

## Conclusions

While presenting the dimensions of effective teaching-learning in clinical education based on a multi-method study, a new framework in relation to epistemological orientations about teaching and learning was developed in this study, based on which each of these dimensions can be conceptualized, and finally, an effective teaching-learning system in clinical medicine education was implemented. In this study, researchers presented a new framework for epistemological beliefs about teaching and learning, based on the framework of Ottenhoff- de Jonge et al. [22] on medical educators' beliefs about teaching, learning, and knowledge. The new framework is a three-dimensional matrix based on which the dimensions of effective teaching-learning in clinical education were explained. Each dimension has a special meaning in terms of epistemological orientation about teaching-learning. Implementing effective teaching-learning in clinical medical education requires moving from the single teaching-centered or learning-centered orientation to the teaching-learning centered orientation. Focusing on the seven dimensions based on the epistemological orientation of teaching-learning is the starting point of effectiveness and improving the quality of clinical education. In order to implement the model developed through this research and the teaching-learning orientation, all the following items should be given serious consideration: selecting motivated students and strengthening their motivation during education; comprehensive development of students during education; training students as lifelong learners; recruitment, employment and retention of competent teachers and their personal and professional development; implementing an educational culture of involving patients in clinical education; use of collaborative teaching-learning strategies; equipping clinical teachers with pedagogical knowledge and motivating clinical teachers and educators to engaging in faculty development courses and programs in medical education, integration of new technologies in medical education (special attention to technology-enhanced clinical education), focus on context and environment and promoting the positive teaching-learning environment, developing educational leadership and management skills of clinical teachers and educators, participative leadership in the clinical environment and funding of medical education.

## Supporting information

**S1 File. Remote qualitative interviews.**
(DOCX)

**S2 File. Expert panel questions.**
(DOCX)

**S3 File. Open-ended questions.**
(DOCX)

## Acknowledgments

The research team are grateful the clinical teachers, medical education specialists, and medical students who participated in this study. Also, we thank and appreciate Dr. Mohammad Palesh, Dr. Mehrnoosh Khoshnoodifar (internal reviewers), Dr. Mitra Amini and Dr. Mohammad-Ali Hosseini (external reviewers); and Dr. Masomeh Kalantarion and Dr. Nasim Gheshlaghi (peer students) for their valuable and useful comments and feedback.

## Author Contributions

**Conceptualization:** Hamed Khani, Soleiman Ahmady, Shirdel Zandi.

**Data curation:** Hamed Khani, Ali Namaki, Shirdel Zandi.

**Formal analysis:** Hamed Khani, Babak Sabet, Ali Namaki, Shirdel Zandi.

**Investigation:** Hamed Khani, Soleiman Ahmady.

**Methodology:** Hamed Khani, Soleiman Ahmady, Babak Sabet, Ali Namaki, Shirdel Zandi, Somayeh Niakan.

**Project administration:** Soleiman Ahmady.

**Resources:** Hamed Khani, Shirdel Zandi.

**Software:** Hamed Khani, Ali Namaki.

**Supervision:** Soleiman Ahmady, Babak Sabet.

**Validation:** Hamed Khani, Ali Namaki, Somayeh Niakan.

**Visualization:** Hamed Khani, Shirdel Zandi.

**Writing – original draft:** Hamed Khani, Shirdel Zandi.

**Writing – review & editing:** Hamed Khani, Soleiman Ahmady, Babak Sabet, Ali Namaki, Somayeh Niakan.

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
