## [Decision Letter · Decision Letter 0]

21 May 2023

PONE-D-23-10842Teaching-learning in clinical education Based on epistemological orientations: A multi-method studyPLOS ONE

Dear Dr. Khani,

Thank you for submitting your manuscript to PLOS ONE. After careful consideration, we feel that it has merit but does not fully meet PLOS ONE’s publication criteria as it currently stands. Therefore, we invite you to submit a revised version of the manuscript that addresses the points raised during the review process.

We look forward to receiving your revised manuscript.

Kind regards,

Ehsan Namaziandost

Academic Editor

PLOS ONE

Journal Requirements:

Reviewers' comments:

Reviewer's Responses to Questions

**Comments to the Author**

1. Is the manuscript technically sound, and do the data support the conclusions?

Reviewer #1: Yes

Reviewer #2: Yes

2. Has the statistical analysis been performed appropriately and rigorously? 

Reviewer #1: Yes

Reviewer #2: N/A

3. Have the authors made all data underlying the findings in their manuscript fully available?

Reviewer #1: Yes

Reviewer #2: Yes

4. Is the manuscript presented in an intelligible fashion and written in standard English?

Reviewer #1: No

Reviewer #2: No

5. Review Comments to the Author

Reviewer #1: Dear author(s),

Having read through your manuscript, I have identified various issues that do not align with the criteria set out for publication in this journal. They have been highlighted in the margin of the manuscript. Please address them point-by-point.

With best regards,

Reviewer #2: Thnak you for the opportunity to review this interesting manuscript. Although I am positive with the study, I still have some commens for improvement. First, the introduction needs to highlight novelty offered in the study. This can be done by reviewing more updated research on the topic. Second, the finidngs should be discussed based on both theoretical perspectives and empirical evidence. I suggest the authors review recent wok on the topic. Laslty, I noted some gramatical erros and typos in the manuscript. Pelase check them and have the mansucript proofead professionally.

6. PLOS authors have the option to publish the peer review history of their article (what does this mean?). If published, this will include your full peer review and any attached files.

Reviewer #1: **Yes: **Afsheen Rezai

Reviewer #2: No

---

## [Author Response · Author response to Decision Letter 0]

18 Jun 2023

Dear Reviewers

Thank you for giving us the opportunity to submit a revised draft of the manuscript "Teaching-learning in Clinical Education Based on Epistemological Orientations: A Multi-method Study" for publication in PLOS ONE journal.

We appreciate the time and effort that you dedicated to providing feedback on our manuscript and are grateful for the insightful comments on and valuable improvements to our paper.

We have carefully reviewed the comments and have revised the manuscript accordingly. Our responses are given in a point-by-point in 'point-by-point response to reviewers file. The changes of the paper have been uploaded in two versions including marked-up (revised manuscript with track changes) and unmarked (revised manuscript without tracked changes).

We hope the revised version is now suitable for publication and look forward to hearing from you in due course.

Kind regards,

Corresponding Author

---

## [Decision Letter · Decision Letter 1]

12 Jul 2023

Teaching-learning in clinical education Based on epistemological orientations: A multi-method study

PONE-D-23-10842R1

Dear Dr. Khani,

We’re pleased to inform you that your manuscript has been judged scientifically suitable for publication and will be formally accepted for publication once it meets all outstanding technical requirements.

Kind regards,

Ehsan Namaziandost

Academic Editor

PLOS ONE

Additional Editor Comments (optional):

Reviewers' comments:

Reviewer's Responses to Questions

**Comments to the Author**

1. If the authors have adequately addressed your comments raised in a previous round of review and you feel that this manuscript is now acceptable for publication, you may indicate that here to bypass the “Comments to the Author” section, enter your conflict of interest statement in the “Confidential to Editor” section, and submit your "Accept" recommendation.

Reviewer #1: (No Response)

Reviewer #2: All comments have been addressed

2. Is the manuscript technically sound, and do the data support the conclusions?

Reviewer #1: Yes

Reviewer #2: Yes

3. Has the statistical analysis been performed appropriately and rigorously? 

Reviewer #1: Yes

Reviewer #2: N/A

4. Have the authors made all data underlying the findings in their manuscript fully available?

Reviewer #1: Yes

Reviewer #2: No

5. Is the manuscript presented in an intelligible fashion and written in standard English?

Reviewer #1: No

Reviewer #2: Yes

6. Review Comments to the Author

Reviewer #1: I thank the authors for considering my comments. The manuscript has improved substantially. However, there are some further minor points that should be considered.

-The discussion part should be more critical in light of the existing literature.

-The contributions of the study for theory and practice should be discussed clearly.

-There are some violations considering in-text citations. They are not in line with the style of PLOSE ONE journal. Please fix them.

-The language of the manuscript needs more improvement. There are a couple of sentence which sound odds. Please proofread the manuscript to increase the readability of the text.

-The references are not following the guidelines of PLOS ONE Journal.

Best,

Reviewer #2: The objective of this research was to develop a conceptual model of effective clinical teaching in undergraduate

medical education and conceptualize its operational framework based on the best fit approach. The revised version has been well written and all my comments are addressed appropriately.

7. PLOS authors have the option to publish the peer review history of their article (what does this mean?). If published, this will include your full peer review and any attached files.

Reviewer #1: **Yes: **Afsheen Rezai

Reviewer #2: No

---

## [Editor Report · Acceptance letter]

9 Nov 2023

PONE-D-23-10842R1 

Teaching-learning in clinical education Based on epistemological orientations: A multi-method study 

Dear Dr. Khani:

I'm pleased to inform you that your manuscript has been deemed suitable for publication in PLOS ONE. Congratulations! Your manuscript is now with our production department. 

Kind regards, 

on behalf of

Dr. Ehsan Namaziandost 

Academic Editor

PLOS ONE